# The Wnt Co-Receptor PTK7/Otk and Its Homolog Otk-2 in Neurogenesis and Patterning

**DOI:** 10.3390/cells13050365

**Published:** 2024-02-20

**Authors:** Qian Hui Tan, Agimaa Otgonbaatar, Prameet Kaur, Angelica Faye Ga, Nathan P. Harmston, Nicholas S. Tolwinski

**Affiliations:** 1Division of Science, Yale-NUS College, Singapore 138527, Singaporeagimaa@u.yale-nus.edu.sg (A.O.); prameet.k87@gmail.com (P.K.); angelicafayega@yale-nus.edu.sg (A.F.G.); harmstonn@cardiff.ac.uk (N.P.H.); 2Program in Cancer and Stem Cell Biology, Duke-NUS Medical School, Singapore 169857, Singapore; 3Molecular Biosciences Division, Cardiff School of Biosciences, Cardiff University, Cardiff CF10 3AX, UK

**Keywords:** Wnt, PTK7, *Drosophila*

## Abstract

Wnt signaling is a highly conserved metazoan pathway that plays a crucial role in cell fate determination and morphogenesis during development. Wnt ligands can induce disparate cellular responses. The exact mechanism behind these different outcomes is not fully understood but may be due to interactions with different receptors on the cell membrane. PTK7/Otk is a transmembrane receptor that is implicated in various developmental and physiological processes including cell polarity, cell migration, and invasion. Here, we examine two roles of Otk-1 and Otk-2 in patterning and neurogenesis. We find that Otk-1 is a positive regulator of signaling and Otk-2 functions as its inhibitor. We propose that PTK7/Otk functions in signaling, cell migration, and polarity contributing to the diversity of cellular responses seen in Wnt-mediated processes.

## 1. Introduction

Signaling pathways such as Wnt regulate cell fate decisions in a tissue-specific manner [1,2]. These processes can be studied in embryonic models as embryonic development involves the coordinated movement of cells along with cell fate choices. Both are aspects of polarity and transcriptional signaling coordinated by the Wnt pathway [3,4].

The Wnt pathway is activated by the binding of Wnt ligand to Frizzled receptor and LRP5/6 co-receptors on the plasma membrane, followed by the recruitment of Disheveled and subsequent activation, or repression of downstream target genes and cellular processes [5]. Canonical Wnt signaling is dependent on β-catenin to transduce signal between the membrane and the nucleus. In the absence of Wnt, β-catenin is phosphorylated and degraded by the destruction complex containing the kinases CK1α and GSK-3 as well as the scaffold proteins Axin and APC. The binding of Wnts to their cognate receptor inactivates the destruction complex and prevents the degradation of β-catenin, allowing its translocation to the nucleus where it regulates expression of target genes via TCF [6,7]. This branch of the pathway is thought to primarily direct transcriptional changes that determine cell fate and proliferation. The non-canonical pathway or β-catenin-independent signaling does not require β-catenin for signal transduction and has mainly been implicated in the regulation of cell organization and polarity [8,9]. Activation of the non-canonical pathway has been found to involve the activation of Disheveled (Dsh in *Drosophila*, Dvl in mammals), followed by Rho GTPase activation and actin rearrangement [5,8,10].

These different branches of the Wnt pathway are activated by the binding of extracellular Wnt ligands to distinct transmembrane receptors and co-receptors, including Frizzled (Fz), Ptk7, and Ror [11,12,13,14,15,16]. Protein Tyrosine Kinase-7 (PTK7 or Off-track 1, Otk-1 in *Drosophila*) and other non-canonical co-receptors have been proposed to direct polarity and migratory behaviors [17,18,19,20]. PTK7 resembles a receptor tyrosine kinase but is predicted to lack kinase activity due to a mutation in the ATP binding site of the putative kinase domain [21].

Studies in *Drosophila* have identified that Ptk7/Otk is a co-receptor for Wnt-4 and Wnt-2 [11,16]. PTK7 functions via two distinct mechanisms: one where the extracellular domain is cleaved and functions as a signal for cell migration on its own, and one where the intracellular domain activates its own signaling pathway [22,23,24,25]. Intracellular Otk signaling during *Drosophila* embryogenesis contributes to axonal pathfinding and epidermal patterning by inhibiting canonical signaling, while activating a non-canonical pathway [11,26,27]. PTK7 affects the canonical Wnt pathway by regulating LRP6 levels in Xenopus [28]. The pathway components downstream of Otk are not fully elucidated but require PTK7 binding to Dsh [10], the recruitment of RACK1 [29], and the protein tyrosine kinase Src which phosphorylates Rho-associated kinase (ROCK) regulating the myosin regulatory light chain to stimulate actomyosin contractility [30].

In *Drosophila*, the best studied Wnt is called Wingless (Wg), but several other homologs show some activity [31,32]. Wnt-4 was originally identified as showing opposing phenotypes to *wg* in patterning and pathfinding functions during neurogenesis [33,34,35,36,37]. Wnts are not thought to contribute to planar cell polarity in *Drosophila* making it unclear what would define a non-canonical Wnt in flies [32,38]. They all bind to Frizzled receptors [39], but some show activity toward other co-receptors, such as Wnt5 and the Derailed receptor [40]. These studies suggest that there are non-canonical Wnt pathways in *Drosophila* possibly using distinct co-receptors [3].

Here, we investigate the mechanism by which Otk-1 transduces and interprets Wnt signals along with its binding partner Otk-2 [16]. We observe Otk-1 expression during embryogenesis in both epithelial as well as neuronal cells. We demonstrate that Otk-1 and its homolog Otk-2 act antagonistically, and that Wnt-4 activates Otk-1 to repress canonical Wnt signaling in *Drosophila* embryos.

## 2. Materials and Methods

### 2.1. Crosses and Expression of UAS Construct

All crosses for transgene expression were performed as previously described [41,42]. For more information, please visit Flybase [43,44]. Expression was driven by daughterless and maternal-GAL4 combinations. All additional stocks were obtained from the Bloomington *Drosophila* Stock Center (NIH P40OD018537). The tandem fluorescent protein timer recombination insertion construct was synthesized by SynbioTech (Monmouth Junction, NJ, USA) and injected by Bestgene (Chino Hills, CA, USA) according to the MiMIC-RMCE protocols [45,46,47], and timer protocols [48,49].

### 2.2. Fly Lines and Crosses

#### 2.2.1. Fly Lines Used

mataTub-Gal4VP16 15 (D. St. Johnston)daughterless-Gal4 BDSC 55850-> Recombined to make mat, da Gal4 (this study)UAS-Otk-1 pUASg 3XHA [11]UAS-Wnt-4 [34,35]UAS-Wnt-4 RNAi BDSC Stock #29442 [50]UAS–Otk-1 RNAi BDSC Stock #25790UAS–Otk-2 RNAi BDSC Stock #38973UAS–Otk-1 RNAi BDSC Stock #55869UAS–Otk-2 RNAi BDSC Stock #55892UAS–Otk-2 RNAi BDSC Stock #57040UAS-Frankenbody HA nanobody-GFP; daughterless-GAL4 [51]Mi[Trojan-GAL4]otk BDSC Stock #76759UAS-wg BDSC 5918 [52]UAS-Td-Tomato BDSC Stock #36328 (Joost Schulte and Katharine Sepp)elavGal4, UAS-GFP BDSC Stock #5146

#### 2.2.2. Fly Crosses Performed

mat, da GAL4 × UAS-Otk-1mat, da GAL4 × UAS-Wgmat, da GAL4 × UAS-Otk-1, UAS-Wnt-4mat, da GAL4 × UAS-Otk-1 RNAimat, da GAL4 × UAS-Otk-2 RNAimat, da GAL4 × UAS-Otk-1 RNAi, UAS-Wnt-4 RNAimat, da GAL4 × UAS-Otk-2 RNAi, UAS-Wnt-4 RNAimat, da GAL4 × UAS-Wnt-4mat, da GAL4 × UAS-Wnt-4 RNAimat, da GAL4 × UAS-Otk-1 RNAi, UAS-Otk-2 RNAida GAL4 × UAS-Otk1, UAS-Franken HA-GFPMi[Trojan-GAL4]otk × UAS-Td-TomatoelavGal4, UAS-GFP × UAS-Otk-1 RNAi, UAS-Otk-2 RNAi

### 2.3. Imaging

#### 2.3.1. Light Sheet Imaging

Otk-1TFP, Otk-1TrojanGal4 > TdTom, and elavGal4 samples were imaged with a Zeiss Lightsheet Z.1 microscope. *Drosophila* embryos were dechorionated using bleach, rinsed twice with water, dried, and loaded into a capillary filled with 1% low-melting agarose Type VII-A in water (Sigma-Aldrich, St. Louis, MO, USA) [53]. 

All light sheet images were acquired with a water immersion objective at W Plan-Apochromat 40×.1.0 UV-VIS detection objective (Carl Zeiss, Jena, Germany). Otk-1TrojanGal4 > TdTom samples were exposed to a 561 nm laser at 10% power every 5 min for 150 cycles. Otk-1Timer samples were imaged using the following settings: 10% laser power for 488 nm, 15% laser power for 561 nm every 5 min for 250 cycles.

Red fluorescent protein (RFP) and superfolder GFP (sfGFP) cluster intensity was quantified using IMARIS 9.0 (Bitplane AG, Belfast, UK). In brief, two time points were extracted, corresponding to the early and late stages of embryo development. All channels were rendered with the same brightness and sensitivity settings in IMARIS, and volume statistics were exported for further analysis in R. Statistical analysis was performed using R (v 4.2.2) and graphed using the ggpubr (v0.6.0) package.

#### 2.3.2. Confocal Imaging 

*Drosophila* embryos were dechorionated using bleach, rinsed twice with water, and immersed in water in an imaging dish. UAS-otk-1 Franken-HA-GFP samples were acquired on the Zeiss LSM 800 (Carl Zeiss, Germany) using 1% laser power for 488 nm. Images were processed using the maximum intensity projection function of the ZEN 2.0 SP software (Carl Zeiss, Germany).

### 2.4. Transcriptomic Analysis

RNA-seq data were assessed for quality using FastQC [54] v0.11.8. Reads were aligned against *Drosophila* genome release BDGP6.22.97 using STAR [55] v2.7.1a and quantified using RSEM [56] v1.3.1. FastQC, STAR, and RSEM results were then compiled and assessed via MultiQC [57] v1.14.

Reads mapping to chrM or annotated as rRNA, Mt_rRNA, snoRNA, or snRNA were removed. Genes with fewer than 10 read mappings on average across all samples were also removed. Read counts were normalized using a regularized log transformation, and principal component analysis was performed to assess sample clustering.

Differential expression analysis was performed using DESeq2 [58]. For pairwise comparisons, log2 fold changes were shrunk using the apeglm [59] algorithm. For clustering analysis, gene-level counts were transformed using a regularized log transformation, converted to z-scores, and clustered using k-means. The average silhouette width of clusters was used to determine the optimal number of clusters (k = 3). For each cluster, Gene Ontology (GO) and KEGG enrichments were performed using clusterProfiler [60] (v4.7.1.003).

## 3. Results

### 3.1. Patterning

Embryonic patterning brings together developmental signaling pathways to impart direction and identity to cells of developing embryos [61]. The Wnt, or Wingless (Wg) pathway in *Drosophila*, is part of anterior to posterior patterning [62,63]. Loss of *wg* leads to a denticle-covered embryo (Figure 1A) while overexpression of *wg* shows a naked or no-denticle phenotype (Figure 1B, compared to normal patterning in Figure 1C) [52]. *Drosophila* Wnt-4 appears to have an antagonistic activity toward *wg* causing extra denticles to appear when overexpressed [34,35]. This function is through a different co-receptor: Off-Track (Otk-1) or Protein Tyrosine Kinase 7 (PTK7) [11]. In *Drosophila*, Otk-1 is expressed in stripes corresponding to the expression of Wnt-4, and both Otk-1 and Wnt-4 oppose Wg signaling [11,27,35]. A recent study suggested that a deletion of both Otk-1 and 2 had no obvious phenotype in embryos so we reexamined the patterning phenotypes using a maternally driven RNAi approach [16].

Otk-1 is not a true tyrosine kinase [29], but instead is likely to act as a scaffold. It interacts with Disheveled (Dsh/Dvl) but does not bind the key β-catenin regulator Axin [11]. We looked at the phenotype of Otk-1 RNAi in embryos through a cuticle assay where a periodic Wnt signal can be observed (Figure 1) [64]. This is an assay used to determine developmental patterning through observation of a simple cell fate choice where cells in the epidermis choose to either make a denticle or not resulting in a segmented/periodic appearance (Figure 1A–C). Loss of Wnt leads to all cells adopting the denticle fate, while Wnt hyperactivation leads to all cells adopting the naked cell fate (compare Figure 1A to Figure 1B). We observed that Otk-1 knockdown led to a loss of some denticles, or a Wnt activation phenotype, but the consistency between segments and across embryos did not show strong penetrance (Figure 1D). Otk-2 knockdown, however, led to a gain of denticle cell fates (Figure 1E) with complete penetrance. The combined loss of both Otk-1 and Otk-2 resulted in a strong loss of Wnt phenotype (Figure 1F) with complete penetrance among embryos that completed cuticle development. These surprising findings suggest that the two Otk proteins play opposing roles in Wnt signaling with Otk-1 being a negative regulator and Otk-2 a positive one. Since Otk-1 and 2 can bind to each other, these findings suggest that Otk-2 is a negative regulator of Otk-1.

We next looked at the relationship of Otk to Wnt-4. We expressed RNAi for Otk-1 and overexpressed Wnt-4. If Otk-1 is the sole receptor for Wnt-4, then it should be epistatic, meaning that the phenotype would be similar to Otk-1 RNAi alone, but this was not the case with embryos showing a complex phenotype with some areas having more denticles and some fewer (Figure 1G) with less than 50% penetrance. We looked at a combination of Otk-1 and Wnt-4 RNAi, but similarly could not make a strong conclusion about epistasis as few embryos made cuticles making patterning difficult to observe (Figure 1H). We also tested the combination of Wnt-4 RNAi and Otk-2 RNAi, which gave a clear finding of loss of patterning similar to loss of Otk-2 alone (Figure 1I) with complete penetrance. As RNAi based knockdown experiments do not provide null conditions for testing epistasis, we can only conclude that the epistasis experiments for Otk-1 were mixed, but Otk-2 RNAi showed consistent behavior acting upstream of Otk-1 and parallel to Wnt-4.

An often-used readout of canonical Wnt signaling during *Drosophila* embryonic development is the expression of the *engrailed* (*en*) gene. Transcription of *en* is activated in anterior to posterior stripes in response to *wg* activity mirroring the activity of the pathway [1,63]. To examine the effect of Otk on *en* expression, we used a combination of two early Gal4 drivers to overexpress and knock down Otk and Wnt-4 in embryos. In wild-type embryos, En protein appears in stripes two to three nuclei thick (Figure 2A). We had previously reported a loss of En protein stripes in response to Otk-1 and Wnt-4 co-expression [11]. Here we showed that overexpression of Wnt-4 (Figure 2B), Wnt-4 RNAi (Figure 2C) and Otk-1 RNAi (Figure 2D) showed little effect on En protein stripes comparable to their mild cuticle phenotypes. We did observe a decrease in both intensity and thickness of En stripes when Otk-2 RNAi was expressed (Figure 2E), and a complete loss of striping when both Otk-1 and 2 RNAi were combined (Figure 2F). UAS-driven overexpression of Otk-1 (Figure 2G) showed little effect on En stripes, while some ectopic En was observed in Otk-1, Wnt-4 double RNAi (Figure 2H). Combining Otk-1 RNAi with Wnt-4 overexpression had little effect on En stripes (Figure 2I). These findings, combined with the patterning phenotypes observed (Figure 1) suggest that Otk-2 and Otk-1 are required for proper patterning by canonical Wnt signaling.

### 3.2. Transcriptomic Analysis Identifies Wnt-4-Dependent and Wnt-4-Independent Clusters 

To further investigate the interaction between Wnt-4, Otk-1, and Otk-2, we collected embryos overexpressing and downregulating these genes in various combinations. Across all conditions, 7939 genes (FDR < 10%) were identified as significantly differentially expressed. K-means clustering of these genes identified three distinct clusters (Figure 3A).

Cluster I (*n* = 1439) contained genes that were activated by Wnt-4 but were unaffected by Otk-1. In other words, this is a Wnt-4-dependent but Otk-1-independent cluster. Within this cluster, we found enrichments for various signaling pathways, including the EGF receptor signaling pathway, MAPK signaling pathway, and Hippo signaling pathway (Figure 3A,B). Cluster II (*n* = 3535) contained genes that were repressed by Wnt-4 and Otk-1. Genes in this cluster were primarily associated with axis specification, canonical Wnt signaling, Hedgehog (Hh) signaling, and Notch signaling. Cluster III (*n* = 2965) contained genes that were activated by Wnt-4 and Otk-1. These genes were primarily involved in cuticle development and neuroactive ligand-receptor interaction. Intriguingly, this cluster also showed enrichment for junction assembly proteins, specifically those involved in apical and septate junction assembly.

#### 3.2.1. Wnt-4 Activates Otk-1 to Oppose Canonical Wnt Signaling

Wnt ligands bind to Frizzled receptors. For downstream signaling to proceed co-receptors are also bound, thus allowing the activation of specific pathways. Wnt-4 has previously been shown to be able to bind Otk-1 in *Drosophila* embryos in a co-immunoprecipitation assay [11]. If Wnt-4 is the ligand for the Otk-1 co-receptor, we would expect Wnt-4 RNAi embryos, Otk-1 RNAi embryos, and Wnt-4 RNAi Otk-1 RNAi embryos to show similar transcriptional profiles, as they would reflect the removal of a ligand and a receptor in combination. As expected, our RNA-seq data show that Wnt-4 RNAi embryos, Otk-1 RNAi embryos, and Wnt-4 RNAi Otk-1 RNAi embryos show similar transcriptional profiles, with an upregulation of Cluster II genes (canonical Wnt signaling) and a downregulation of genes in Cluster III (junction assembly). Wnt-4 opposes canonical Wnt signaling in *Drosophila* embryos [34]. Consistent with these results, we observe an upregulation of canonical Wnt signaling in Wnt-4 RNAi embryos, Otk-1 RNAi embryos, and Wnt-4 RNAi Otk-1 RNAi embryos (Figure 3B), suggesting that Wnt-4 signals through Otk-1 to inhibit canonical Wnt signaling.

#### 3.2.2. Wnt-4 Activates Otk-1 to Affect Tight Junction Assembly

Wnt-4 RNAi embryos, Otk-1 RNAi embryos, and Wnt-4 RNAi Otk-1 RNAi embryos showed downregulation in Cluster III genes, which are enriched for tight junction assembly. The transmembrane adhesion regulators *Mesh* and *Gliotactin* were strongly downregulated (Appendix A). Downregulation of these genes has been shown to induce mislocalization of Lethal giant larvae (Lgl), Discs Large (Dlg), Coracle (Cora), and Fasciclin 3 (FasIII) proteins, inhibiting septate junction formation [65,66]. Otk-1 may play a role in the formation of septate junctions, and further studies could examine this angle in greater detail.

#### 3.2.3. Otk-2 Is an Inhibitor of Otk-1

To further investigate the relationship between Otk-1 and Otk-2, we compared the transcriptional profiles of UAS-Otk-1, Otk-1 RNAi, and Otk-2 RNAi embryos. Otk-1 binds to Otk-2, and it has been suggested that Otk-2 functions redundantly with Otk-1 [16]. Consistent with previous studies, we show that Otk-1 RNAi embryos and Otk-1, Otk-2 double RNAi embryos have similar transcriptional profiles (Figure 3A), with an upregulation of canonical Wnt signaling (Cluster II) and a downregulation of tight junction assembly genes (Cluster III).

Surprisingly, Otk-2 RNAi embryos show a dramatically distinct transcriptional profile from Otk-1 RNAi embryos and are instead more similar to UAS-otk-1 embryos. In Otk-2 RNAi embryos, canonical Wnt signaling is downregulated (cluster II) while tight junction assembly genes are upregulated (cluster III). This is the direct opposite of Otk-1 RNAi embryos, which have increased canonical Wnt signaling and decreased expression of tight junction assembly genes. Thus, we propose that Otk-2 functions as an inhibitor of Otk-1. In Otk-2 RNAi embryos, downregulating Otk-2 reduces Otk-1 inhibition, resulting in more Otk-1 available for signaling. This is similar to UAS-Otk-1 embryos, where increased expression of Otk-1 results in higher levels of Otk-1 available for signaling. Thus, we conclude that Otk-2 is an inhibitor of Otk-1 since downregulating Otk-2 activates similar transcriptional profiles as overexpressing Otk-1 in embryos.

Combining RNAi for both Otk isoforms suggests that Otk-2 functions upstream of Otk-1. Transcriptional profiles for the double Otk-1 and Otk-2 RNAi embryos were similar to Otk-1 RNAi alone, indicating that Otk-1 is epistatic to Otk-2. In the absence of Otk-1, any reduction in Otk-2 could not alleviate the reduced signaling caused by the lack of Otk-1. As opposed to the removal of Otk-2 alone in Otk-2 RNAi embryos which results in greatly reduced inhibition of Otk-1, causing a sharp increase in Otk-1 signaling, explaining why Otk-2 RNAi embryos had similar transcriptional profiles to UAS-Otk-1. Our results suggest that Otk-2 is an inhibitor of Otk-1, rather than a redundant co-receptor.

These results led to a working model for the Wnt-4-Otk-1-Otk-2 axis. The Wnt-4 ligand binds to co-receptor Otk-1, repressing canonical Wnt signaling and activating pathways that modify tight junction assembly. Otk-2 functions as an inhibitor of Otk-1, possibly acting competitively with Wnt-4 to bind Otk-1.

### 3.3. Tracking Otk Expression

The two previously proposed roles for Otk in patterning and neurogenesis [11,26,67] were confirmed by transcriptomic analysis (Figure 3B,C). This suggested that Otk must be functioning at different times in both epithelial and neuronal tissues. Studies to ascertain the expression patterns of Otk-1 and 2 have shown that they are coexpressed in most tissues [16]. These studies used fixed embryos with antibody and RNA in situ approaches, so to test these findings in vivo we applied new approaches using fluorescent proteins. Using a Trojan Gal4, where a Gal4 coding exon is inserted into the Otk-1 gene [68], should reproduce the expression pattern of the endogenous promoter. We crossed the Otk-1 TrojanGal4 flies to fluorescent proteins to observe expression in embryos. (Figure 4 and Appendix A). We observed epithelial, striped expression in mid-stage embryos (Figure 4A,A’), and strong neuronal expression in later stages (Figure 4B,B’). This recapitulates most of the antibody and in situ studies showing some early epidermal expression and predominantly neuronal expression in later stages. We did not extend these studies to Otk-2 due to a lack of easily targeted sites such as the ones designed for Otk-1, but according to published expression analysis, Otk-2 appears to be expressed in a nearly identical pattern to Otk-2 leading to the inference that Otk-2 expression would look similar in these experiments [16].

In a second approach, we used Recombination-Mediated Cassette Exchange (RMCE) using a MiMIC insertion in the Otk-1 locus to insert two fluorescent protein tags into the coding sequence. This approach is known as the tandem fluorescent protein timer approach to measure protein stability over time. Essentially, a fast-folding super folder GFP is usually observed within 20 min of expression, while the slowly folding RFP protein can take up to 4 hours [48,49,69]. As expected, this approach yielded much less robust expression of fluorescent proteins as compared to the Trojan-Gal4 approach but did reveal a specific, highly expressing tissue (Figure 5, Appendix A) which appears to be the Stomatogastric Nervous System and Ring Gland [70].

To image the localization of overexpressed Otk-1, we used the frankenbody approach [71]. This uses overexpression of an α-HA nanobody fused to GFP or mCherry, which can be co-expressed with any HA-tagged transgene for live imaging. We looked at the localization of Otk-1-HA (Figure 6A, Appendix A). This approach allowed us to observe membrane localization of Otk-1 in embryonic epithelial tissues. This expression mirrored that observed in fixed embryos expressing Otk-1-HA (Figure 6B).

As expression in the nervous system does not necessarily mean function, we looked to confirm previously published neuronal phenotypes [26,37,67], challenged by the newer deletion study [16], by expressing RNAi for both Otk-1 and Otk-2 specifically in neurons using the neuronal driver embryonic lethal abnormal vision (elavGal4 [72,73]). Normal embryos develop a nervous system as they mature into larvae (Figure 6C, Appendix A). When we expressed RNAi for both Otk-1 and Otk-2 in neurons, the nervous system failed to form completely (Figure 6D, Appendix A). This finding is surprising as the name Off-track comes from pathfinding problems in *otk-1* mutants but does point to the function that Otk-2 plays in Otk-1 regulation with much stronger phenotypes observed when Otk-2 is knocked down.

## 4. Discussion

PTK7 is of interest in the cancer biology field as it is often found dysregulated in cancer. Its function has never been entirely clear with functions proposed in polarity and signaling [17,18,19,20]. In vertebrates, it functions as a component of planar cell polarity (PCP) pathways and binds to Wnts. In *Drosophila*, PCP is significantly different from vertebrates with no secreted Wnt required [32]. Vertebrate PCP focuses mainly on cell movements such as convergent extension which is a cell migration process. Recent findings in *Drosophila* and vertebrates have linked PTK7 to intestinal regeneration and cellular senescence [25,74]. In this model, PTK7 is responsible for providing a cue for the migration of intestinal stem cells toward the site of injury regenerating the epithelium. This type of cell migration is more similar to vertebrate PCP and explains why PTK7 and another Wnt co-receptor Ror do not show fly PCP phenotypes but rather participate in cell migration processes such as neuronal pathfinding and stem cell migration [75,76,77,78].

Our study specifically investigated the function of Otk in embryogenesis. There have been conflicting reports published on the function of Otk with a recent publication showing that a deletion of both Otk-1 and Otk-2 showed no embryonic phenotypes [16]. Our previous work had shown patterning phenotypes, especially with overexpressed Otk-1 and Wnt-4 [11]. One major issue that affects embryonic patterning studies is the presence of maternally deposited mRNA which means that for most components a germline clone strategy is required [79,80]. The original allele of Otk turned out to be required in oogenesis or acquired a secondary mutation on the chromosome [81], so could not be used in this manner. The Otk-1/Otk-2 deletion was sterile and therefore could not directly address the contribution of maternal mRNA. In this study, we instead turned to RNAi with early expressing maternal Gal4 drivers to address the effect of Otk in patterning. By using this approach, we find that Otk-2 knockdown causes major patterning and neuronal phenotypes in embryos. Loss or overexpression of Otk-1 by itself has mild to no phenotype but can synergize with Otk-2 to cause a complete loss of patterning.

Our analysis supports our previous findings that Otk signaling primarily acts to repress the canonical branch of the Wnt pathway whilst activating a non-canonical branch. This contrasts with studies that have suggested that Ptk7/Otk is involved in β-catenin regulation rather than the non-canonical pathway [82]. Other studies have proposed that Ptk7/Otk is not the co-receptor for Wnt-4 but is bound by different members of the Wnt family, i.e., Wnt-2 [16]. Our analysis here did not look at the physical interactions previously reported [11], but by comparing transcriptional profiles, we show that the effect of perturbing Wnt-4 and Otk is highly similar providing evidence that Otk is a co-receptor for Wnt-4 in *Drosophila* embryos. Linnemannstöns et al. only observed a phenotype when both Otk-1 and Otk-2 were knocked out in flies and proposed that these two genes function redundantly in the pathway. However, our analysis of the transcriptional effects of Otk-1 RNAi and Otk-2 RNAi revealed that these are orthogonal to each other (Figure 3A).

We provide a possible model for the regulation of Otk signals (Figure 7). We show that Otk-2 functions as a repressor of Otk-1, a function that has not yet been discovered in the mammalian system. Our approach used knockdown rather than deletion possibly explaining why phenotypes were observed. Changing levels of the components provided a titratable system to observe signaling changes supporting work done in the gut model, focusing on the intracellular signals rather than the secreted extracellular domain [25].

Our transcriptional profiles point to two major embryonic processes: patterning and neurogenesis. In patterning, we observe perturbed signaling pathways leading to the loss of pattern phenotypes. In particular, the Erk and Wnt pathways were affected. We did not pursue the effects of Otk-1 on neurogenesis in detail as the original pathfinding phenotypes were not directly challenged [16]. We did however observe that the expression of Otk-1 and Otk-2 RNAi showed severe disruption of neurogenesis suggesting that expression in neurons is important and confirming the transcriptional analysis findings. One very exciting finding was that both the Hippo pathway and cell adhesion were affected. Both relate to cell migration and could help explain the findings that Otk has no phenotype in classical *Drosophila* PCP but does function in cell migration-related polarity.

Despite much work, the downstream effectors of polarity pathways remain poorly defined. PTK7 interacts with Src suggesting a possible mechanism for junctional changes downstream [83]. PTK7 is a pseudo-kinase so cannot be activated by the standard RTK transphosphorylaton mechanism, but a second kinase such as Src or CK1ε which phosphorylate Ror could be involved [30]. Although in this study, we did not focus on the downstream pathway, the effect of Otk on adhesion and Hippo pathway does suggest that the proposed mechanism of activation through Src and ROCK is likely to be correct [30]. Src is a famous oncogene [84] contributing to the invasive phenotype, or metastatic potential of cancer cells, one of the hallmarks of cancer [85]. PTK7 may contribute to this explaining its association with a wide range of cancers. We observed transcriptional effects on signaling pathways, but most interesting was the finding that apical basal polarity and Hippo pathways were affected. This effect could explain the cancer and stem cell association and will have to be explored in a cell migration and invasion model [25,86].

## Figures and Tables

**Figure 1 cells-13-00365-f001:**
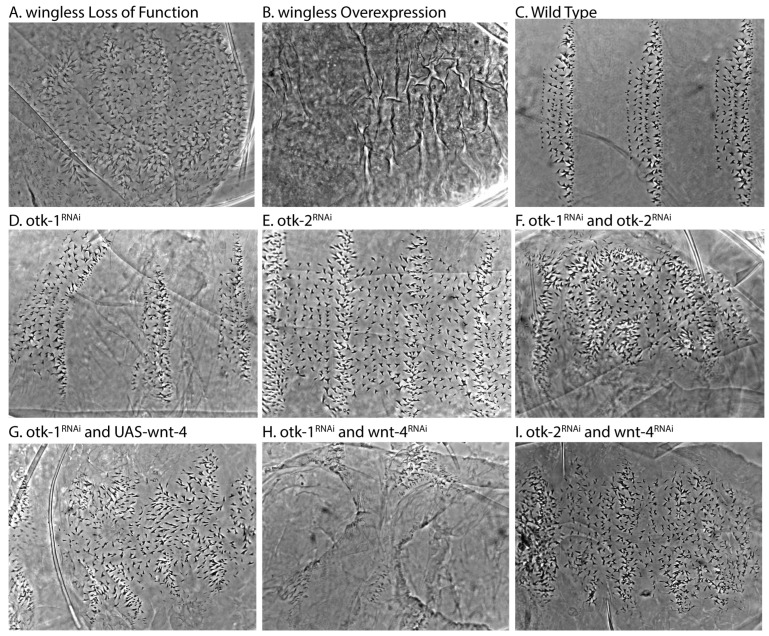
Otk and Wnt-4 in embryonic patterning. (**A**) Loss of segment polarity patterning in a *wg^IG22^* embryo or denticle lawn phenotype. (**B**) Loss of segment polarity or naked phenotype due to *wg* overexpression, as compared to wild-type segment polarity patterning (**C**). (**D**) *otk-1^RNAi^* results in denticle pattern disruption but not a clear segment polarity phenotype. (**E**) *otk-2^RNAi^* showed a disruption of segment polarity, with normally naked cuticles covered in denticles. (**F**) Co-expression of both *otk-1^RNAi^* and *otk-2^RNAi^* led to a more severe denticle lawn phenotype. The combination of *otk-1^RNAi^* and *wnt-4* overexpression (**G**) led to a slight increase in denticles over *otk-1^RNAi^* alone. The combination of *otk-1^RNAi^* and *wnt-4^RNAi^* showed a loss of cuticle (**H**), making the observation of patterning phenotypes difficult, but the combination of *otk-2^RNAi^* and *wnt-4^RNAi^* showed a near denticle lawn phenotype (**I**). For RNAi experiments, we examined embryos n > 100 per experiment.

**Figure 2 cells-13-00365-f002:**
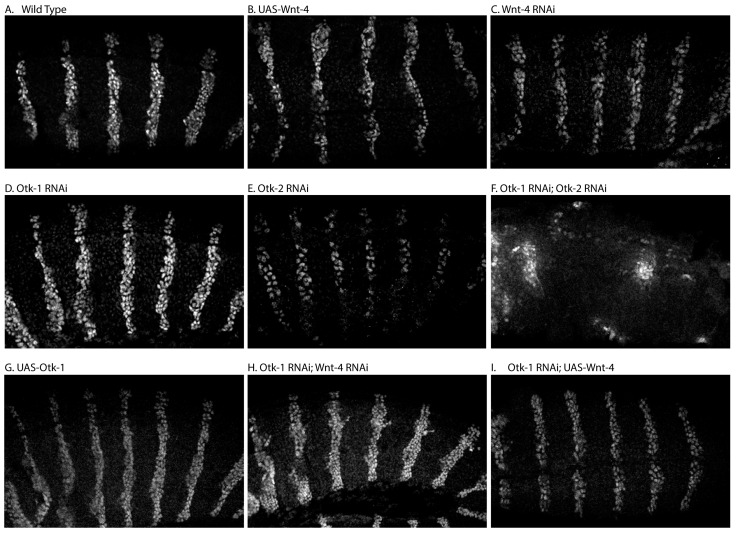
Effect of Otk on Engrailed expression. (**A**) Wild-type embryos show a striped pattern of En protein expression usually in two to three cell nuclei per stripe. (**B**) Overexpression of *wnt-4* showed a small effect on stripe width. (**C**) *wnt-4^RNAi^* did not appear to affect En protein. (**D**) *otk-1^RNAi^* showed a very mild effect on stripe width. (**E**) *otk-2^RNAi^* showed lower levels of En expression and loss of En-positive cells. (**F**) Co-expression of both *otk-1^RNAi^* and *otk-2^RNAi^* led to a complete disruption in the En expression pattern. Overexpression of Otk-1 showed a mild to no effect on En expression (**G**). The combination of *otk-1^RNAi^* and *wnt-4^RNAi^* showed some extopic En-positive cells (**H**). The combination of *otk-1^RNAi^* and *wnt-4* overexpression led to a slight decrease in En levels (**I**). For each condition, we examined at least three individual embryos of a similar stage.

**Figure 3 cells-13-00365-f003:**
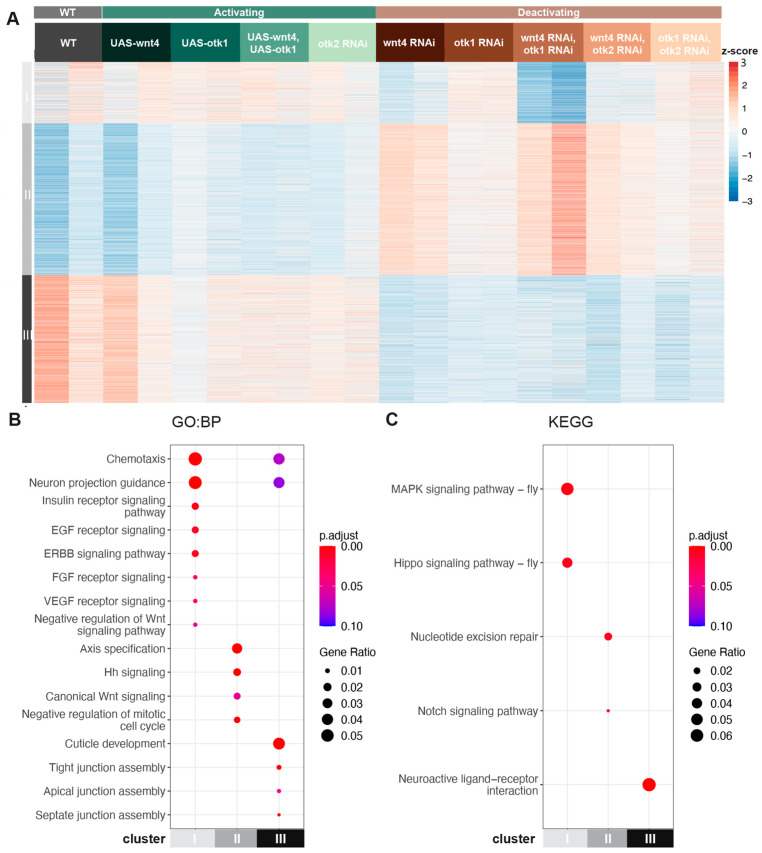
Gene expression changes in *Drosophila* embryos downstream of Wnt-4 and Otk-1. (**A**) Heatmap of gene expression in embryos expressing various levels of Wnt-4, Otk-1, and Otk-2. Clusters I and II are Wnt-4 dependent, while Cluster III is Wnt-4 independent. (**B**) Gene Ontology (GO) enrichment analysis highlighting key signaling and metabolic pathways associated with each cluster (FDR < 10%). Downregulation of Wnt-4 results in reduced EGF, and VEGF signaling (Cluster I) and increased canonical Wnt and Hedgehog signaling (Cluster II). Downregulation of Wnt-4 and Otk-1 results in decreased tight junction assembly (Cluster III). (**C**) KEGG pathway enrichment analysis highlighting key signaling and metabolic pathways associated with each cluster (FDR < 10%). Downregulation of Wnt-4 results in reduced MAPK, Hippo signaling (Cluster I), and increased Notch signaling (Cluster II).

**Figure 4 cells-13-00365-f004:**
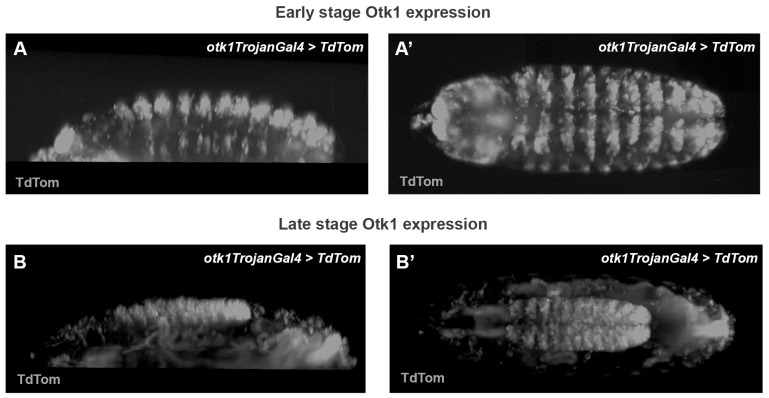
Otk-1 TrojanGal4 in *Drosophila* embryos. (**A**,**A’**) Still images from live imaging showing expression of Otk-1 in ventral, epithelial stripes during mid stages of embryogenesis. (**B**,**B’**) Still images from Lightsheet imaging, showing later stage expression of Otk-1 in neural tissue. Images and videos are representative of three independent experiments.

**Figure 5 cells-13-00365-f005:**
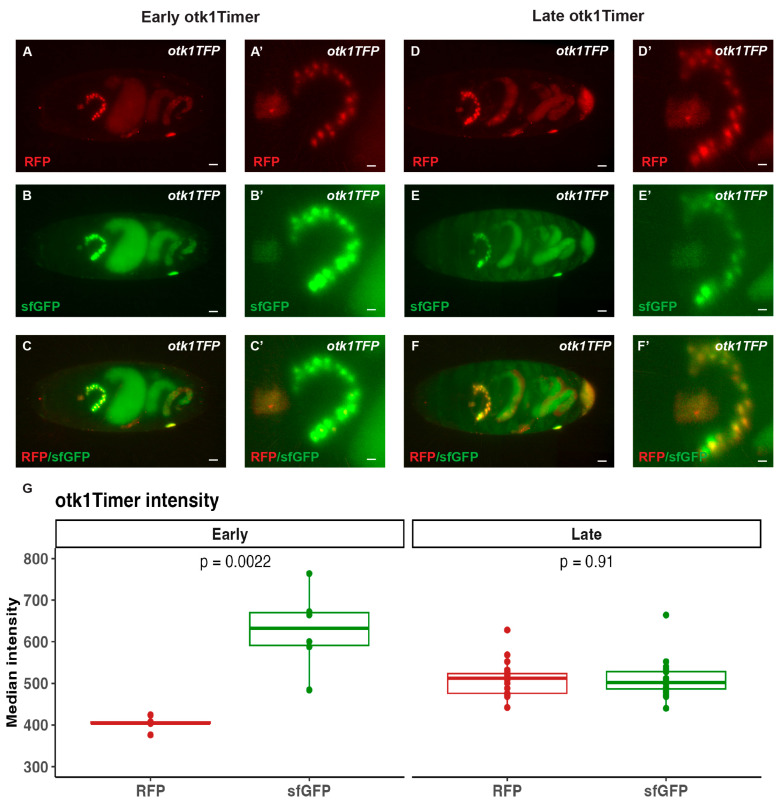
Still images from Otk-1, with two fluorescent proteins inserted into the locus, lightsheet imaging in *Drosophila* embryos. A rapidly folding fluorescent protein (super folder GFP) and a slow folding fluorescent protein (RFP) were inserted directly into the *otk-1* locus. (**A**,**A’**) RFP signal was only observable in the stomatogastric cells in mid embryonic stages along with a strong sfGFP signal (**B**,**B’**) which overlapped (**C**,**C’**). In late embryonic stages both sfGFP and RFP were observed in stomatogastric cells (**D**–**F**,**D’**–**F**’). (**G**) Quantification of otk1Timer intensity in the ring gland for early and late-stage embryos. Median intensity was rendered via IMARIS. Statistical *p*-value calculated using unpaired *t*-test in R. Images and videos are representative of three independent experiments.

**Figure 6 cells-13-00365-f006:**
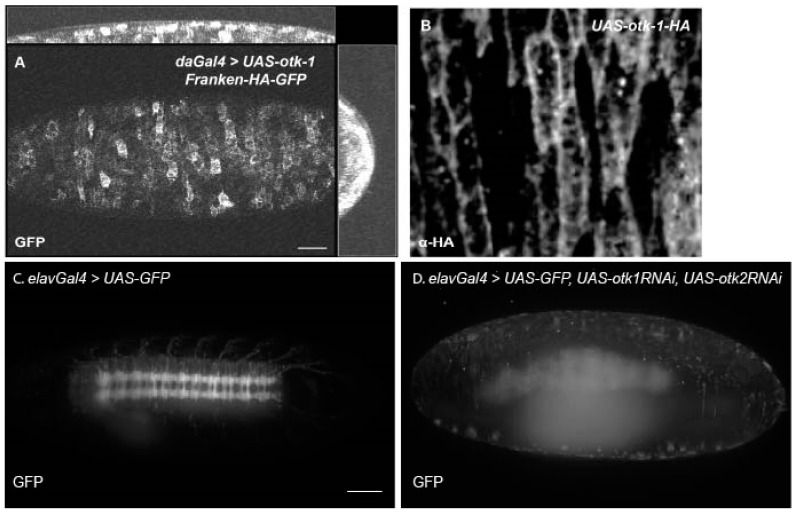
Overexpression of Otk-1 in embryos. (**A**) Still image from confocal microscopy live imaging of HA-tagged otk-1 imaged using αHA frankenbody fused to GFP shows an irregular pattern of overexpression at the membrane of epithelial cells. (**B**) Membrane localization and irregular levels were confirmed by fixed, αHA antibody staining. (**C**) Still image from lightsheet movie showing embryonic neural development using neuronally driven GFP. (**D**) Addition of RNAi for both Otks causes loss of GFP signal and disorganized neurogenesis. Images and videos are representative of three independent experiments.

**Figure 7 cells-13-00365-f007:**
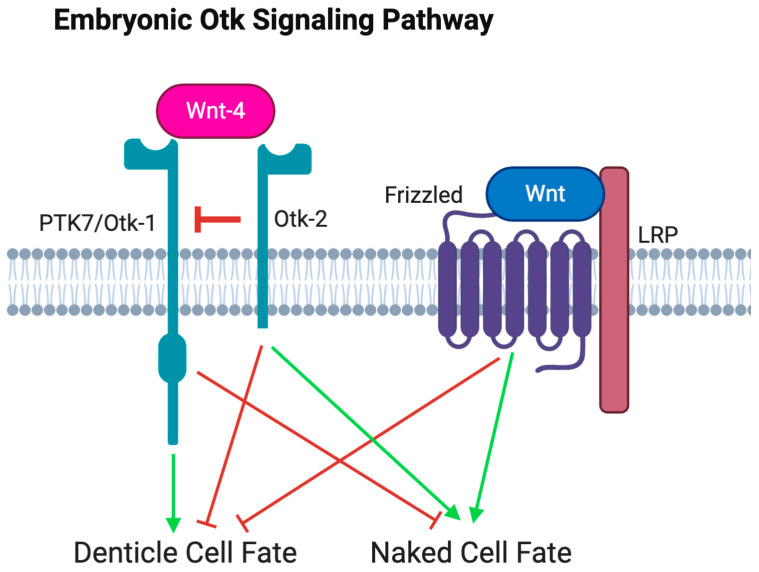
Model of Otk Signaling. Canonical Wnt signaling determines the naked cell fate. Otk-2 represses Otk-1 function to prevent denticle cell fate activation. This repression can be overcome by the presence of the Wnt-4 ligand.

## Data Availability

RNA seq data have been deposited on GEO and can be accessed at the following accession number: GSE245408. Reproducible code for RNA-seq analysis is available at https://github.com/harmstonlab/otkwnt_embryo.

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
