# Peer review of "The Wnt Co-Receptor PTK7/Otk and Its Homolog Otk-2 in Neurogenesis and Patterning"

_cells, 2024, doi:10.3390/cells13050365_

Round 1

Reviewer 1 Report

Comments and Suggestions for Authors

PTK7/otk modulate classical and non-canonical wnt signaling and are therefore of great interest to researchers of a wide range of fields from developmental to cancer biology among others. Despite several studies during the past 10 to 15 years, the published literature contains inconsistencies which may be due to difference between species, cell types and experimental approaches. 

Tan et al. address a specific inconsistency, namely a function of otk1 and otk2 in embryonic segmentation during AP patterning. In addition, the authors provide genetic data both by cuticle phenotype and transcriptional profiling about functional relationship of otk1, otk2 and wnt4. Lastly, the authors provide data about the expression pattern in vivo for otk1 and otk2 employing fluorescent reporters. 

As the data are conclusive and largely convincing, I strongly recommend to publish the manuscript. 

I am listing a few recommendations that will improve the study, especially with respect to inconsistencies in literature mentioned above. 

-       The cuticle phenotype are clear and convincing. A statement or numbers about the penetrance and severance of the phenotypes would make the data more convincing than only exemplary images. 

-       The modulation of wnt signaling as shown by the cuticle phenotypes may be also shown on a molecular level by en staining, for example. Such a staining would make the cuticle phenotypes more convincing and remove counter arguments towards a role of otks in wnt signaling. 

-       A shortcoming of the PlosGenet paper from 2014 from the Wodarz group is the neglect of a maternal contribution for otk1 and otk2. The authors may provide any information (preferably from literature or published expression data) that otk1 and otk2 RNA or proteins are present in mature eggs. Do the reports show any maternal expression?

-       Some of the figure captions contain interpretation of the data. Description and interpretation of the presented data should be in the text not captions. 

-       Monochromic images are better visible in grey scale than in a colorful LUT e. g. Fig.3, 4, 5). 

-       Please provide a literature reference to the Bloomington stock center and FLYBASE. I assume that the authors have used Flybase as a reference for genetic nomenclature, at least. 

-       A paragraph explaining and discussing the inconsistencies in the literature and the issue of maternal and zygotic phenotypes would be useful for readers not aware of the controversy. 

Author Response

Reviewer 1:

PTK7/otk modulate classical and non-canonical wnt signaling and are therefore of great interest to researchers of a wide range of fields from developmental to cancer biology among others. Despite several studies during the past 10 to 15 years, the published literature contains inconsistencies which may be due to difference between species, cell types and experimental approaches. 

Tan et al. address a specific inconsistency, namely a function of otk1 and otk2 in embryonic segmentation during AP patterning. In addition, the authors provide genetic data both by cuticle phenotype and transcriptional profiling about functional relationship of otk1, otk2 and wnt4. Lastly, the authors provide data about the expression pattern in vivo for otk1 and otk2 employing fluorescent reporters. 

As the data are conclusive and largely convincing, I strongly recommend publishing the manuscript. 

I am listing a few recommendations that will improve the study, especially with respect to inconsistencies in literature mentioned above. 

-       The cuticle phenotype are clear and convincing. A statement or numbers about the penetrance and severance of the phenotypes would make the data more convincing than only exemplary images. 

We have added N numbers to the phenotypes observed with penetrance estimates. This was an oversight in the original manuscript. For all conditions, as these were overexpression experiments, we screened at least 100 embryos. We have indicated the penetrance for the phenotypes, especially for Otk-2 as this was complete.

-       The modulation of wnt signaling as shown by the cuticle phenotypes may be also shown on a molecular level by en staining, for example. Such a staining would make the cuticle phenotypes more convincing and remove counter arguments towards a role of otks in wnt signaling. 

Again, we should have done this initially. We have now added En staining for 9 conditions to enhance the Wnt phenotype.

-       A shortcoming of the PlosGenet paper from 2014 from the Wodarz group is the neglect of a maternal contribution for otk1 and otk2. The authors may provide any information (preferably from literature or published expression data) that otk1 and otk2 RNA or proteins are present in mature eggs. Do the reports show any maternal expression?

Yes, indeed there is maternal expression. One of the limitations of our old study published in 2011 was that the original mutant did not allow for germline clone mutation as no eggs were produced. We have added this to the paper in the discussion. The deletion used by the Wodarz group was sterile so did not address this issue.

-       Some of the figure captions contain interpretation of the data. Description and interpretation of the presented data should be in the text not captions. 

This has now been fixed.

-       Monochromic images are better visible in grey scale than in a colorful LUT e. g. Fig.3, 4, 5). 

We have converted the images to grey scale where only one color was used.

-       Please provide a literature reference to the Bloomington stock center and FLYBASE. I assume that the authors have used Flybase as a reference for genetic nomenclature, at least. 

We have added these, apologies for the oversight. Bloomington grant is in the acknowledgements, and the flybase citations are included.

-       A paragraph explaining and discussing the inconsistencies in the literature and the issue of maternal and zygotic phenotypes would be useful for readers not aware of the controversy. 

We have significantly expanded the discussion to explain the inconsistencies. We were initially uncertain how to address the inconsistencies, but we have done so now.

Reviewer 2 Report

Comments and Suggestions for Authors

This work examines the patterning role of Wnt co-receptors Otk-1 and Otk-2 in Drosophila. The authors show that Wnt4 interacts with Otk-1 to inhibit canonical Wnt signaling. However, Otk-2 may function as an inhibitor of Otk-1. Although this study is of some interest, there are several important issues that need to be addressed before the manuscript can be considered for publication.

1.   The introduction section does not provide sufficient information on Ptk7, Otk-1 and Otk-2 in canonical and non-canonical Wnt signaling.

2.   In the analyses of functional interactions between Wnt4, Otk-1 and Otk-2 in transcriptional regulation (section 3.2), there is no experimental validation of representative up- and down-regulated genes.

3.   In addition to the repression of genes associated with specification and different signaling pathways, Wnt4 and Otk-1 also activate the expression of genes involved in apical and septate junction assembly. The authors should further discuss how these changes in gene expression are related to patterning or other processes.

4.   Otk-1 is not a tyrosine kinase and likely acts as a scaffold. How does Wnt4 activate Otk-1 to inhibit canonical Wnt signaling? The authors should provide an explanation on this observation.

5.   There is no functional study of Otk-1 and Otk-2 in neurogenesis. In the descriptive study of Otk excpression, the authors show neural expression of Otk-1 at late stages of development but they do not provide any data on Otk-2 expression.

6.   The discussion section is particularly disappointing. Possible mechanisms underlying the antagonistic interactions between Wnt4/Otk-1 and canonical Wnt signaling or Otk-2 are not discussed. There is also no discussion on the regulation of canonical and non-canonical Wnt signaling by Otk-1 and Otk-2.

Comments on the Quality of English Language

The manuscript needs proof reading.

Author Response

Reviewer 2:

This work examines the patterning role of Wnt co-receptors Otk-1 and Otk-2 in Drosophila. The authors show that Wnt4 interacts with Otk-1 to inhibit canonical Wnt signaling. However, Otk-2 may function as an inhibitor of Otk-1. Although this study is of some interest, there are several important issues that need to be addressed before the manuscript can be considered for publication.

  1. The introduction section does not provide sufficient information on Ptk7, Otk-1 and Otk-2 in canonical and non-canonical Wnt signaling.

We have expanded the introduction to include more details.

  1. In the analyses of functional interactions between Wnt4, Otk-1 and Otk-2 in transcriptional regulation (section 3.2), there is no experimental validation of representative up- and down-regulated genes.

We did not go into details looking at specific genes in this manuscript. We were focused on showing the Wnt phenotype and the effect of Otk-2.

  1. In addition to the repression of genes associated with specification and different signaling pathways, Wnt4 and Otk-1 also activate the expression of genes involved in apical and septate junction assembly. The authors should further discuss how these changes in gene expression are related to patterning or other processes.

We have significantly expanded the discussion section. The adhesion finding is the most exciting for future work!

  1. Otk-1 is not a tyrosine kinase and likely acts as a scaffold. How does Wnt4 activate Otk-1 to inhibit canonical Wnt signaling? The authors should provide an explanation on this observation.

If only we knew. There have been several proposed interacting partners but there isn’t a definitive mechanism known. By far the most interesting is the Src connections as this brings together signaling and adhesion, and cancer and metastasis. We are developing tools to look at this, but this data is far too preliminary.

  1. There is no functional study of Otk-1 and Otk-2 in neurogenesis. In the descriptive study of Otk excpression, the authors show neural expression of Otk-1 at late stages of development but they do not provide any data on Otk-2 expression.

We did not pursue neurogenesis in detail in the original study as these previously published studies were not disputed by Wodarz and colleagues directly. As Otk-2 has not been well studied, the molecular tools for in vivo observation do not exist. But, Wodarz observed identical expression of Otk1 and Otk2 in all tissues tested suggesting that transcription is co-regulated, meaning that they have identical expression patterns. We added an experiment to address the reviewer’s question where we expressed Otk1 and 2 RNAi specifically in neurons. This abolished neurogenesis!  We hope this adds the functional significance to the localization and transcriptional analysis findings.

  1. The discussion section is particularly disappointing. Possible mechanisms underlying the antagonistic interactions between Wnt4/Otk-1 and canonical Wnt signaling or Otk-2 are not discussed. There is also no discussion on the regulation of canonical and non-canonical Wnt signaling by Otk-1 and Otk-2.

We have substantially expanded the discussion to address this. We have added a model figure as well.

Round 2

Reviewer 1 Report

Comments and Suggestions for Authors

I support publication.

Reviewer 2 Report

Comments and Suggestions for Authors

Issues raised in the previous round of review have been addressed.